# Navigation Between Alzheimer’s Disease (AD) and Its Various Pathophysiological Trajectories: The Pathogenic Link to Neuroimmunology—Genetics and Neuroinflammation

**DOI:** 10.3390/ijms26178253

**Published:** 2025-08-26

**Authors:** Abdalla Bowirrat, Albert Pinhasov, Aia Bowirrat, Rajendra Badgaiyan

**Affiliations:** 1Adelson School of Medicine, Department of Molecular Biology, Ariel University, Ariel 40700, Israel; albertpi@ariel.ac.il; 2Department of Psychiatry, Texas Tech University Health Science Center, Midland, TX 79430, USA; badgaiyan@gmail.com; 3Department of Orthopedic Surgery, Hasharon Hospital, Rabin Medical Center, Tel Aviv University, Petah-Tikva 40700, Israel; 216bow@gmail.com

**Keywords:** Alzheimer’s disease, blood–brain barrier, APOE4, neuroinflammation, metabolic syndrome, brain insulin resistance

## Abstract

One hundred and eighteen years have passed since Alzheimer’s disease (AD) was first diagnosed by Alois Alzheimer as a multifactorial and complex neurodegenerative disorder with psychiatric components. It is inaugurated by a cascade of events initiating from amnesic-type memory impairment leading to the gradual loss of cognitive and executive capacities. Pathologically, there is overwhelming evidence that clumps of misfolded amyloid-β (Aβ) and hyperphosphorylated tau protein aggregate in the brain. These pathological processes lead to neuronal loss, brain atrophy, and gliosis culminating in neurodegeneration and fueling AD. Thus, at a basic level, abnormality in the brain’s protein function is observed, causing disruption in the brain network and loss of neural connectivity. Nevertheless, AD is an aging disorder caused by a combination of age-related changes and genetic and environmental factors that affect the brain over time. Its mysterious pathology seems not to be limited to senile plaques (Aβ) and neurofibrillary tangles (tau), but to a plethora of substantial and biological processes, which have also emerged in its pathogenesis, such as a breakdown of the blood–brain barrier (BBB), patients carrying the gene variant APOE4, and the immuno-senescence of the immune system. Furthermore, type 2 diabetes (T2DM) and metabolic syndrome (MS) have also been observed to be early markers that may provoke pathogenic pathways that lead to or aggravate AD progression and pathology. There are numerous substantial AD features that require more understanding, such as chronic neuroinflammation, decreased glucose utilization and energy metabolism, as well as brain insulin resistance (IR). Herein, we aim to broaden our understanding and to connect the dots of the multiple comorbidities and their cumulative synergistic effects on BBB dysfunction and AD pathology. We shed light on the path-physiological modifications in the cerebral vasculature that may contribute to AD pathology and cognitive decline prior to clinically detectable changes in amyloid-beta (Aβ) and tau pathology, diagnostic biomarkers of AD, neuroimmune involvement, and the role of APOE4 allele and AD–IR pathogenic link—the shared genetics and metabolomic biomarkers between AD and IR disorders. Investment in future research brings us closer to knowing the pathogenesis of AD and paves the way to building prevention and treatment strategies.

## 1. Introduction

Alzheimer’s disease (AD) is a complex neurodegenerative aging brain pathology that was first described by the German psychiatrist and neuropathologist Alois Alzheimer [1,2] in 1906 as a chronic multifaceted illness characterized by an episodic and amnesic-type memory impairment, impoverishment of language and visuospatial deficits, loss of cognitive and executive abilities, attention, and affect, mood changes, apathy, and increased dependence on others, which is a brain problem affecting the elderly [3]. AD is a predominant and incurable chronic debilitating disorder, more than 60–80% of all types of dementias [4]. AD and other forms of dementia present substantial global public health challenges. As the inevitable consequence of the transformation of the population structure, population aging has emerged as a core issue and has become a global public health challenge for human society in the 21st century. The proportion of the global population aged 65 years and older is projected to nearly double from 8.5% in 2015 to 16.7% in 2050 [5,6].

Currently, approximately 55 million individuals worldwide are afflicted with dementia, with approximately 35 million solely affected by AD. This condition ranks as the seventh leading cause of death globally and constitutes a major public health concern that profoundly influences the well-being of populations worldwide. Globally, the incidence of dementia is steadily escalating, rising from 38.35 × 10^6^ (95% UI: 26.42 × 10^6^–52.05 × 10^6^) cases in 1990 to 98.37 × 10^6^ cases in 2021 (95% UI: 67.13 × 10^6^–134.25 × 106), and the Average Annual Percent Change (AAPC) was 0.06 (95% CI: 0.05–0.07). However, although the number of deaths attributed to dementia increased from 6.63 × 10^6^ in 1990 (95% UI: 1.58 × 10^6^–18.70 × 10^6^) to 19.53 × 10^6^ in 2021 (95% UI: 4.86 × 10^6^–52.00 × 10^6^), the age-standardized mortality rate did not undergo a significant alteration (the AAPC = 0; 95% CI: −0.02 to 0.03). Globally, the disability-adjusted life years (DALYs) among those aged 65 and older linked to dementia grew by 176% between 1990 and 2021, from 11.77 million to 32.55 million [7], and the DALYs rates globally ascended from 135.72 × 10^6^ (95% UI: 62.18 × 10^6^–303.65 × 10^6^) in 1990 to 363.33 × 10^6^ (95% UI: 166.04 × 10^6^–800.36 × 10^6^), but the AAPC was merely 0.03 (95% CI: −0.01 to 0.05), suggesting that the change was insignificant [5].

The neuropathological hallmarks of AD include overwhelming evidence of clumps of misfolded amyloid-β (Aβ—a 36–43 amino acid peptides) and hyperphosphorylated tau proteins, which aggregate in the brain. The first leads to the formation of extracellular amyloid-beta (Aβ) deposition that forms neurotic plaques, and the second leads to the formation of intracellular neurofibrillary tangles. In the prodromal stage of the disease, the pathological process is characterized by abnormal protein processing leading to the aggregation and accumulation of Aβ peptides, aberrant activation of the brain’s innate immune system cells, and neurotoxicity [8,9,10,11,12,13,14,15,16].

Nevertheless, AD pathology is not limited to the accumulation of the hallmark senile plaques (Aβ) and the aggregation of the hyperphosphorylated microtubule-associated protein tau into neurofibrillary tangles (NFTs) in the brain. Other biological processes are also involved in AD pathogenesis, such as blood–brain barrier (BBB) neurovascular dysfunction, which is considered one of the relevant pathophysiological domains involved in the AD pathogenesis framework as a potential player and may play a vital role in the beginning and progression of AD [17,18,19,20,21,22].

According to the two-hit vascular hypothesis of AD [23], any cerebrovascular damage/(BBB dysfunction) could lead to a chain of events, represented by the accumulation of amyloid-β (Aβ) within the brain. It is well-known that one of the essential functions of the BBB is regulation of the mechanism of clearance of amyloid-β (Aβ) across the barrier. Thus, a decrease in the clearance abilities of the BBB may promote the built-up of Aβ plaques in extracellular matrix. This considered an initial insult, which itself is sufficient to initiate neuronal loss and neurodegeneration [24,25,26,27]. BBB damage may also induce neurodegenerative processes via the penetration of neurotoxic substances through it into the brain [28], lead to neuroinflammation [29], and provoke pericyte-mediated cerebral hypoperfusion [30]. In addition, far from the prevailing line of thought or most accepted theories, such as the cholinergic theory, the Aβ cascade hypothesis, or the abnormally excessive phosphorylated tau protein, there are other opinions on what may play a role in the etiology of AD that depend more on central factors, which may underpin the pathogenesis of AD and other dementias. Herein, we discuss and draw attention to various events that may have a crucial role in the pathogenesis of AD (Figure 1, Figure 2 and Figure 3): “Impairment of the immune system (immune-senescence) may be implicated deeply in the pathogenesis of AD” [31,32,33,34,35]; environmental factors; and type 2 diabetes (T2DM) and metabolic syndrome (MS) have also emerged as early markers that may provoke some pathogenic pathways that lead to or aggravate AD progression and pathology. Additionally, there are numerous AD features that require further understanding such as the role of the gene variant APOE4, chronic neuroinflammation, decreased glucose utilization and energy metabolism, as well as brain insulin resistance (IR) [36,37,38,39].

## 2. Pathogenic Link Between the Blood–Brain Barrier (BBB), APOE4 Polymorphism, and Alzheimer’s Disease (AD)

The blood–brain barrier (BBB) is created during early embryonic angiogenesis, approximately at the second embryonic week (W2), by angioblasts forming a vascular plexus of immature blood vessels, which progressively branches to produce a vascularized brain [40]. It is a widened continuous tightly sealed monolayer of brain that contains non-fenestrated endothelial cell membrane within brain micro-vessels. Brain circulation is sophisticated; for example, brain capillaries are a key site of the BBB, and their length in the human brain is about 650 km, which accounts for >85% of the total cerebral blood vessel length, providing the largest endothelial surface area for solute transport exchanges between the blood and brain and vice versa (e.g., ~120 cm^2^/g of brain [41,42]. The mean distance between the BBB and neurons is ~8 μm, allowing the rapid diffusion of molecules across the brain interstitial space from capillaries to neurons and vice versa [42]. The multi-functionality of the BBB shows its importance in the healthy and diseased brain; physiologically, it regulates the influx and efflux of biological substances essential for the brain’s metabolic activity and for neuronal survival. Indeed [23,43], the BBB is a selective gatekeeper for the brain as a semipermeable distinctive and chemical barrier, which plays a meticulous physiological role in the regulation of paracellular permeability, ion balance, and nutrient transport, and it performs the hemodynamic orchestration of the cerebral blood flow (CBF) to meet the metabolic demands of the neurons [40,44,45]. Thus, the BBB ensures a stable brain internal milieu by maintaining homeostasis and an adequate microenvironment, which is essential for adequate synaptic and neural activity [46]. Moreover, the BBB represents a biological barrier, through the interface between the neural tissues of the CNS and blood components, including circulating cells of the immune system; it guarantees the export of potentially neurotoxic blood-derived debris from the brain to the bloodstream, removes microbial pathogens, and expels red blood cells and leukocytes. Additionally, due to its barrier role, it protects the brain from exogenous components and xenobiotics, and filters the blood flow [47,48]. The particularity of the BBB as a semipermeable barrier of the brain has aroused the interest of many researchers; Paul Ehrlich (1885, 1906) first examined this phenomenon. He noticed that a peripherally infused dye did not stain the brain tissue [49]. This finding was further supported by a later observation published over 110 years ago by his associate Edwin E. Goldmann who applied the same trypan dyes to the cerebrospinal fluid [50]. The dye stained only the brain tissue without extravasating in the periphery [51]. These illustrations support the role of the BBB as an intricate barrier that strictly regulates the passage of substances between the blood and brain parenchyma selectively [50].

Anatomically, the BBB is a highly specialized multicellular structure formed by brain microvascular endothelial cells (BMVECs), which are characterized by a lack of fenestrations, the presence of sealed junctions, and minimal pinocytotic activity; these, along with astrocyte end-feet unsheathing the capillary and pericytes embedded in the capillary basement membrane (BM), form a functional element: the neurovascular unit (NVU) [41,44,52,53], which comprise highly specialized cells. They cover all brain microcapillaries with a total surface of between 12 and 18 m^2^ in an adult human brain [54]. The brain microvascular system comprising small blood vessels (diameter < 20 µm) maintains the blood supply, transmits signals, and transfers information between astrocytes, microglia, and neurons [55].

Additionally, BMVECs show an uninterrupted compact junction, exhibiting remarkably diminished pinocytotic activity relative to the microvascular endothelial cells found in peripheral organs. The complex tight junction inhibits the paracellular diffusion of water-soluble molecules into the CNS. In addition, it expresses important transporters and efflux pumps on their surface, such as ATP binding cassette (ABC), and synthesizes relevant neurotransmitter molecules, such as nitric oxide (NO) [40].

Furthermore, the restricted expression levels of endothelial adhesion molecules on the surface of BMECs strictly control immune cell trafficking into the CNS [56]. Due to its lipophilic nature, hydrophobic compounds and gases can diffuse across the BBB by passive diffusion, but larger and hydrophilic compounds require specific transporters located within the barrier. Only solutes of a molecular weight below 400 Daltons (Da) can circulate freely through the BBB endothelium [57]. Typically, 98% of small-molecule drugs and close to 100% of large-molecule drugs fail to enter the central nervous system (CNS) through the tight barrier of endothelial cells [58].

Due to its role as a unique physical and biochemical barrier that is a combination of a physical transporter and metabolic barrier and its multi-functionality, the BBB is a specific physiological compartment that strictly regulates the passage of substances between the blood and brain parenchyma. There are two further membrane compartments in the brain that form a barrier between the blood and cerebrospinal fluid (CSF): the arachnoid epithelium, forming the middle layer of the meninges, and the choroid plexus epithelium [59].

Under pathological conditions, the level of expression of endothelial adhesion molecules, such as intercellular adhesion molecule-1 (ICAM-1) and vascular cell adhesion molecule-1 (VCAM-1), is upregulated and is critical for immune cell infiltration into the CNS [56]. The BBB’s dynamics and functional instability during neuroinflammation—which is the response of the CNS to endogenous and/or exogenous factors that disrupt normal cellular homeostasis, lead to severe consequences, involving both a loss of the BBB’s selective transport mechanisms and a reduction in its structural integrity [60,61]. It was shown that changes in the BBB’s permeability are implicated in the pathophysiology of neurodegenerative diseases, including AD, by facilitating neuroinflammation through unregulated protein entry [23,62]. Moreover, because AD is an aging-related disease characterized by systemic abnormalities in both intracellular and extracellular microenvironments for almost all organs [63], the age-related decrease in pericyte cells further corrupts the BBB, impairing the blood flow and neuronal function [63]. We believe that risk factors such as neuroinflammation and ischemic stroke can lead to the initial breakdown and/or destruction of the architectural integrity of the BBB, leading to the subsequent leakage of the BBB and cerebral edema with concurrent neuronal atrophy, early miscommunication, and deprivation of endothelial cell-to-cell connections. Although, the initial damage to the BBB may eventually be repaired with the restoration of endothelial tight junctions (TJs) [64,65,66], which is an essential molecular component of the BBB composed of claudins, occludins, zonula occludens-1 (ZO-1), ZO-2, ZO-3, and adhesion molecules [67,68]. It seems that the activation of inflammatory pathways as well as other toxic processes may be the primary malefactors that cause brain damage, shortly after BBB dysfunction. However, even a low degree of chronic BBB injury may lead to the BBB’s breakdown and physiological changes including the opening of the tight junctions (TJs), differential expression of protein transport, and the loss of organization of the neurovascular unit (NVU) network, leading to the entry of cells and molecules not usually found in the brain [40]. These effects have been identified in neuroinflammatory conditions associated with neurodegenerative diseases and CNS infections, as well as with inflammation typical in healthy aging [20,40]. The breakdown of the BBB in AD involves both a loss of selective transport mechanisms and a reduction in the structural integrity, which often precedes detectable AD symptomatology and neurophysiological changes [24,69,70]. Dysfunction of the BBB is supported by the anatomical thickness and functional changes in the cerebral microvasculature, which might be directly responsible for the pathogenesis of sporadic AD, or indirectly, or at least could be a consequence of the disease process itself that participates synergistically with other pathogenic mechanisms in the development of neurodegeneration [71].

This fact has been confirmed by more than 20 independent post-mortem human studies. These studies have shown increased brain capillary leakages and perivascular accumulation of blood-derived fibrinogen, thrombin, albumin, and immunoglobulin G (IgG), loss of BBB tight junctions, red blood cells’ (RBC) extravasation, impaired glucose transport, impaired P-glycoprotein-1 function, perivascular deposits of blood-derived products, cellular infiltration, microbial pathogens associated with the degeneration of endothelial and pericytes—cells nestled in the wall of cerebral capillaries, and endothelial cells [72,73,74]. All these events create inflammation and activate the innate immune responses, which can initiate multiple pathways of neurodegeneration.

## 3. Pathogenic Link Between Apolipoproteins Isoforms in AD and Their Role in the BBB’s Breakdown

The apolipoprotein E (APOE) gene belongs to a family of fat-binding proteins (apolipoproteins), located on chromosome 19, which encodes and binds to a specific liver and peripheral cell receptor. It is essential for the normal catabolism of triglyceride-rich lipoprotein constituents. In the CNS, APOE is expressed mainly in astrocytes and microglia, and in peripheral tissues, it is primarily produced by the liver and macrophages [75,76]. APOE encodes a major lipid-carrier protein in the brain [77], as well as vascular mural cells and choroid plexus cells. APOE modulates multiple pathways: its activities are associated with the endocytosis of lipoproteins, synaptic plasticity, membrane integrity, neurogenesis and neuronal degeneration, neuroinflammation, mitochondrial function, tau phosphorylation, and Aβ metabolism [78]. There are three isoforms of APOE in humans: APOE2, APOE3, and APOE4, and there are six different genotypes, i.e., three homozygous and three heterozygous [79]. These isoforms have functional and structural differences, inconsistencies, and discrepancies in their interaction with low-density lipoprotein (LDL) receptors [80].

As with almost all genes, individuals carry two allele copies of APOE, which can be either homozygous (APOE2/APOE2, APOE3/APOE3, APOE4/APOE4) or heterozygous forms (APOE genotypes series are involved in cholesterol metabolism and immune modulation) [81]. Within the cell, APOE plays a role in cellular processes and a prominent role in the overall health of neurons, involved in the maintenance of the cytoskeleton, mitochondria, and dendrites [80]. In the circulation, APOE presents as part of several classes of lipoprotein particles, including chylomicron, very low-density lipoprotein, low-density lipoprotein, and some high-density lipoprotein. Moreover, APOE has a crucial role in amyloid-beta protein (Aβ) clearance, aggregation, and deposition [82,83]. The main associated pathological isoform of APOE in AD is the APOE-ε4 genotype; it is the highest risk factor for late-onset Alzheimer’s disease (LOAD), with the underlying mechanism of this link being both presynaptic and postsynaptic dysfunction [84]. The APOE ε4 gene variant promotes Aβ plaque formation [85], which facilitates the loss of key presynaptic proteins [86], disrupts long-term potentiation and plasticity [87], leads to a reduction in the dendritic density [88], and has a role in the hereditary pathogenesis of AD—up to 4-fold in people in the heterozygous form (APOE3/APOE4 or APOE2/APOE4) and up to 15-fold in the homozygous form (APOE4/APOE4) [69,77,89]. It is worth mentioning, particularly in the case of AD, that association alone does not mean causation, and not every individual who carries APOE ε4 will develop the disease. However, AD is a multifactorial disorder that generally requires the involvement of environmental risk factors accompanied by genetic factors to cause the disease [90,91]. Destructive environmental factors orchestrate together with genetic risk factors to stimulate and accelerate the emergence of the disease. Not every individual carrier of two APOE4 copies is necessarily at risk, but their fate to develop AD is inevitable [91,92]. The presence of APOE ε4 accelerates the age of onset of AD by roughly 15 years in carriers compared to non-carriers (mean age of onset in those who are carriers being 68.4 years versus 84.3 years in non-carriers [37]. In this context, it was shown that individuals who were cognitively intact and carried either one or two copies of APOE4 had a leaky BBB, initially in the hippocampus and in the para-hippocampal gyrus [93]. Remarkable atrophy in these two regions, due to BBB dysfunction, was observed, leading to memory and cognitive impairment. Moreover, histological analysis of post-mortem brain tissue has reported that the BBB’s breakdown in AD patients reduced cerebral blood flow, neural loss, and behavioral deficits independent of Aβ, and it was more noticeable among APOE4 carriers compared to those with APOE3 or APOE2 [94,95,96,97,98].

To cast more light on the association of APOE4 with the BBB, Montagne et al. [93] demonstrated that the APOE4 isoform is secreted by pericytes cells, located close to endothelial cells that line cerebral capillaries at the BBB. APOE4 activates an array of proteins, beginning with the protein cyclophilin A (CypA) in the pericytes, which, in turn, stimulates a downstream signaling pathway involving activation of the inflammatory protein matrix metalloproteinase-9 (MMP9) in pericytes and in endothelial cells. This activation by APOE4 leads to MMP-9–mediated degradation of the BBB’s tight junction and basement membrane proteins causing the BBB’s breakdown [93,99,100,101]. Controversially, APOE3 and APOE2 but not APOE4 act via low-density lipoprotein receptor–related protein-1 (LRP1) on pericytes to inhibit the pro-inflammatory cyclophilin A (CypA)–matrix metallopeptidase-9 (MMP-9) pathway. On other hand, it is well-known that pericytes normally safeguard the BBB [102,103] by preventing the breakdown of the tight junctions located between endothelial cells. In support of this idea, a biomarker of pericyte injury—a soluble form of a protein known as platelet-derived growth factor-receptor-β (sPDGFRβ)—in addition to cyclophilin A (CypA) and to matrix metalloproteinase-9 MMP9) proteins—which are part of the inflammatory pathway—are implicated in APOE4-driven pericyte damage and the BBB’s breakdown, and all are elevated in the CSF of APOE4 carriers [98,104]. Thus, APOE4-status is a risk factor for the BBB’s breakdown via activation of the Cyp-A-MMP9 pathway [99,105] and has been associated with increased hippocampal BBB leakiness and higher sPDGFRβ [93], the novel and sensitive biomarker of BBB disfunction [106,107].

Due to the fact that pericytes are adjacent to the capillary endothelial cells and thereby part of the neurovascular unit, they can, through constriction, regulate capillary blood flow and clean Aβ out of the brain; thus, they are crucial for maintaining the BBB’s overall integrity. Sagare and colleagues [108] have shown that pericytes shed sPDGFRβ into the CSF in response to noxious stimuli. sPDGFRβ may thereby serve as a biomarker of pericyte degeneration and a proxy for the BBB’s integrity [109,110,111,112,113]. Consequently, sPDGFRβ, is increased in AD [114], and it is linked to APOE4-status [93]; other studies reported an association with cognitive dysfunction irrespective of AD pathology [109]. On the other hand, Cicognola and colleagues identified age-dependent effects on sPDGFRβ and associations with neuroinflammation but no association with the AD biomarkers, APOE4, or cognitive decline [115]. Notwithstanding, a routine method used in clinical practice to measure the integrity of the BBB was studied by Halliday et al. [105], who demonstrated that the ratio of albumin in the cerebrospinal fluid (CSF)/plasma albumin quotient (Q_Alb_) is an established marker of the BBB’s breakdown [116].

However, the low molecular weight of albumin (66.5 kDA [117]) raises the question of the appropriateness of this method to detect minor paracellular BBB leakage. Kurz et al. [118] limited the diagnostic sensitivity of CSF/Q_Alb_ to thin BBB changes in the context of AD with some research illustrating increased QAlb in patients with dementia [17,119] but not in mild cognitive impairment (MCI) [119,120,121].

It is well-known that the liver is the single organ capable of synthetizing albumin, and no active transport mechanisms across the BBB have been described. Nevertheless, of all these inconsistencies and reservations, it is still a suitable candidate to assess the BBB’s integrity [26,122].

The Qalb index is considered a biomarker of the BBB. It is the ratio of albumin levels in the CSF and blood and is an indicator of the serum Alb leakage into the CNS [123]. It was shown that AD patients have increased Qalb [123], which correlates with the progression of cognitive impairment [17,119]. Indeed, the CSF/serum albumin (Qalb) index was introduced as a reliable measure, with values of >9 indicating the BBB’s dysfunction [123]. Pathological Qalb was described ranging from 16% [123] to 22% in patients with mild to moderate AD dementia [17,119,124]. Importantly, increases in Q_Alb_ values correlated positively with both CypA and active MMP-9 CSF levels in all the studied individuals (r = 0.37, p < 0.01; r = 0.45, p < 0.01) indicating that the greater the increase in CypA and active MMP-9 levels, the greater the magnitude of the BBB’s breakdown assayed by Q_Alb_. It was seen that APOE4 transgenic model mice showed an increase in BBB vulnerability [125] and the CSF concentration of CypA-MMP9 in APOE4 but not in APOE3 and APOE2 transgenic mice; this finding also occurs in humans [43,100,112].

## 4. Pathogenic Link Between Insulin Resistance (IR), Alzheimer’s Disease (AD), and Blood–Brain Barrier (BBB) Disorders

Since the discovery and isolation of insulin in 1921 by Frederik Banting and Charles Best of Toronto University, this hormone has received great attention, initially due to its role in regulating peripheral glucose levels. Later, there was more research on the reciprocal relationship between brain and insulin signaling, followed by understanding the mechanism of transport of insulin into the brain from the circulation, as well as how insulin enters the brain by binding brain micro-vessels via the blood–brain barrier (BBB) [126].

Insulin is a peptide hormone produced by the β-cells located in the pancreatic islets of Langerhans, which vary in size from 50 to 300 μm in diameter and contain a few hundred to a few thousand endocrine cells. The islets of Langerhans (named endocrine pancreas) are separated anatomically and functionally from pancreatic exocrine tissue (which secretes pancreatic enzymes and fluid directly into ducts that drain into the duodenum). In healthy humans, the number of islets is about one million, weighing 1–2 g and representing 1 to 2 percent of the total mass of the pancreas. At least 70 percent of islets of Langerhans are β-cells that secrete insulin, which are mostly localized in the core of the islet. The β-cells are surrounded by other three major cell groups: alpha cells that secrete glucagon, delta cells that secrete somatostatin, and PP cells that secrete pancreatic polypeptide, as well as by other two minor cell types (D_1_ and enterochromaffin cells) that produce hormones and synthesize serotonin, respectively [127].

The islets of Langerhans were named to honor the German physician Paul Langerhans (1847–1888), who first discovered the islets in 1869.

Insulin is composed of alpha (with 21 amino acids) and beta chains (with 30 amino acids), which are linked together by sulfur atoms [127].

In 1921, Frederick Grant Banting (1891–1941) and Charles H. Best (1899–1978) were the first to isolate insulin in pancreatic extracts. They were followed by Nicolas C. Paulescu (1869–1931), who called the substance “pancrein”. The purified extract was obtained with the collaboration of Scottish physiologist J.J.R. Macleod (1876–1935). Banting and Macleod shared the 1923 Nobel Prize for physiology or medicine for their discovery.

Today, the role of insulin in the regulation of glucose metabolism in peripheral tissue, lipid metabolism, vascular regulation, and cell growth is well-known [128], but the quandary of insulin’s multi-functionality in the brain still needs more clarification. A plethora of research studies conducted in humans and laboratory animals have shed light on its various activities and indicate that it motivates nutritional metabolism and cerebral bioenergetics and increases synaptic efficiency (by enhancing the capacity of a presynaptic input to influence postsynaptic output); increases synaptic viability by strengthening and maintaining active synapses through increased expression of cytoskeletal and extracellular matrix elements and postsynaptic scaffold proteins; and increases dendritic spine formation, and turnover of neurotransmitters, such as dopamine. Insulin also has a role in the proteostasis process that regulates and stabilizes proteins within a cell. It plays an important role in the pathogenesis of AD by influencing the clearance of amyloid-β peptides and enhancing the phosphorylation of tau, which are hallmarks of AD [129,130].

Insulin is needed for the optimal activity of the brain. It helps to maintain the nerve action potential, neuronal ion gradients, cell membrane lipid remodeling, signal conduction, and other pleiotropic biological effects.

Indeed, the brain requires a continuous flow and large quantity of energy to maintain proper activity. It is worth noting that the brain constitutes only 2% of the total body mass, but it consumes around 25% of the glucose and 20% of the oxygen used by the whole body [131,132,133,134].

Intriguingly, glucose is an obligate fuel in the CNS, responsible for cerebral energy metabolism and utilized to maintain neuronal activity via oxidative metabolism both at rest and in activated states. This is illustrated by the disproportionate metabolic rate of the brain relative to most of the other organ systems. In fact, the brain prefers glucose as a source of energy (the brain requires 6–7 mg/100 g of glucose per minute, which is equivalent to 120–130 g per day) [135]. Because glucose is a polar substance, it cannot pass freely through the cell membrane. It requires a special transporter, the glucose transporter (GLUT), to enter a cell. The primary glucose transporters in the brain are GLUT1 and GLUT3, while in the peripheral circulation, GLUT4 transports glucose into the cells [136,137,138]. Even though the GLUT4 level is relatively low in the brain, it is important to maintain neuronal function because it is responsible for glucose influx into synaptic areas, especially during high synaptic activity [139].

The delivery of energy sources into the brain occurs through a sophisticated passage, the Blood–Brain Barrier (BBB), which permits the passage of lipid-soluble substances. Other substances need transporters, including glucose [140]. After the entrance of glucose through the BBB, the glucose links to neurons through two pathways. The first is via the interstitial fluid, where glucose connects to the GLUT1 transporter, spreads into the interstitial fluid, and then enters neurons through GLUT3 [141]. Second is the astrocytes pathway: after passing through the GLUT1 transporter of vascular endothelial cells, it then enters astrocytes through GLUT1, and in astrocytes, glucose is converted to glycogen for storage or to lactic acid by glycolysis. Lactic acid leaves the astrocytes to the extracellular matrix by MCT1 (monocarboxylate transporter-1) or MCT4 (monocarboxylate transporter-4) and enters the neurons by MCT2 (monocarboxylate transporter-2). This mechanism is called the Astrocyte–Neuron Lactate Shuttle Model [142]. During the phenomenon of insulin resistance, the insulin pancreatic production increases to meet the demand of chronically elevated levels of glucose in the circulation and/or the increased amount of adipose tissue that requires insulin for its glucose metabolism [143]. The consequences of insulin resistance (IR) come from the fact that insulin binding to its receptors diminishes dramatically, and the glucose transport into cells is remarkably affected, causing the decreased bio-utility and bioactivity of insulin in the target organs [144,145]. Thus, an inverse relationship exists between insulin resistance and the blood glucose level. As the resistance increases, the glycemic regulation of insulin decreases, leading to a hyperglycemic state [146].

Insulin resistance is the primary pathophysiology of diabetes mellitus, and it is considered a reason for obesity, metabolic syndrome, and different cardiovascular diseases. Brain insulin resistance is the process in which the quantity of insulin in the brain diminished or the brain cells do not respond to insulin [147,148]. However, because the brain does not act as a storehouse of glucose in a hyperglycemic state, insulin resistance does not obligatorily provoke a decrease in glucose concentration in the brain tissue, but it may affect the synaptic activity [128].

The underpinning pathogenesis of insulin resistance and Alzheimer’s disease (AD) still needs more clarification. Both meet the common pathologic exclusiveness, encompassing amyloid genesis, bioenergetic dysfunction, inflammation, and obesity, which together strengthen the notion that insulin resistance may accelerate the appearance of AD [149,150]. One of the mechanisms by which insulin impacts cognitive abilities is by affecting the cerebral energy metabolism. Insulin resistance (IR) has catastrophic consequences on brain function [149,150].

For example, the role of insulin deficiency and insulin resistance (IR) in AD has emerged over the past three decades as a potential candidate for the pathogenesis of AD [151,152,153]. There are a number of pathways that link IR with AD [154,155].

Initially, in the case of an IR state, three detrimental events occur; the IR is followed by compensated peripheral hyperinsulinemia and the resultant hyperglycemia or glucose intolerance. Definitively, IR is observed among individuals with impaired insulin-stimulated glucose output into adipocytes tissues and muscle, accompanied by impaired insulin suppression of hepatic glucose output [156]. This phenomenon of reduced cells’ response to insulin/insulin resistance that leads to hyperinsulinemia can occur due to genetic polymorphisms of tyrosine phosphorylation of the insulin receptor, insulin receptor proteins, and PIP-3 kinase, or abnormalities of the GLUT 4 function, and/or environmental factors [157,158,159,160].

Insulin resistance is a complicated pathophysiological disorder with an impaired biologic response of the target tissues to insulin stimulation and an impaired ability to inhibit glucose production and stimulate peripheral glucose elimination, often with hyperinsulinemia to maintain normoglycemia [161].

The etiology of IR includes factors that disturb the insulin signaling pathway and decrease the peripheral target tissue’s responsiveness to insulin, abnormalities in receptor binding, autophagy, and intestinal microecology, in addition to metabolic dysfunction of the liver and other abnormalities in the host extracellular environment such as lipo-toxicity, inflammation, hypoxia, and immunity abnormalities that can trigger intracellular stress factors in key metabolic target tissues, which impairs the normal metabolic activity of insulin in these tissues thereby provoking the progression of whole-body IR [162,163].

When IR develops, compensatory hyperinsulinemia occurs due to the increased secretion of insulin from the pancreatic β-cell to achieve normoglycemia, which leads to an inadequate or vicious cycle of IR ↔ hyperinsulinemia [154,155,158,164,165,166].

This cycle of IR–hyperinsulinemia causes metabolic consequences including hyperglycemia, high blood pressure, hyperuricemia, dyslipidemia, high levels of elevated inflammatory markers, endothelial dysfunction, and cardiovascular diseases, and may lead to metabolic syndrome and type 2 diabetes. All these consequences may be implicated in AD pathogenesis to different degrees [158,165,166,167].

Chronic elevation in peripheral insulin (peripheral hyperinsulinemia) levels impacts central insulin availability and function. Indeed, peripheral hyperinsulinemia leads to an increase in the insulin level in the brain, because the transport of molecules across the blood–brain barrier (BBB) is highly affected by the variation in their peripheral levels, especially the high level of insulin [149,167,168].

Insulin is degraded into the brain by the insulin-degrading enzyme, (IDE, also called insulysin). In humans, the gene encoding IDE is located on the long arm of chromosome 10 (q23-q25) and contain 24 exons and a large sequence of introns [169]. It presents as an atypical spiral shape structure, confirming its unique enigmatic enzymological properties. IDE is a polypeptide with a molecular weight of 110 kDa, and it is a neutral thiol-dependent metallopeptidase, bound to the metal Zn^2+^ [170]. Originally known as the main enzyme involved in the cleavage of insulin as well as other amyloidogenic peptides, such as the β-amyloid (Aβ) peptide, it eliminates Aβ’s neurotoxic effects—one of the hallmarks of Alzheimer’s disease (AD)—which shows the relationship between IDE, diabetes, and AD [171,172].

The IDE has, therefore, been long envisaged as a potential therapeutic option for metabolic and neurodegenerative diseases [173]. It cleaves many peptides unevenly, including β-amyloid, demonstrating a critical role in pathophysiological processes regulated by these peptides [174,175,176,177,178,179]. It is well-known that the insulin-degrading enzyme (IDE) represents the link and the key factor in the crosstalk between hyperinsulinemia and AD [180]. Several epidemiological studies have demonstrated that hyperinsulinemia and type 2 diabetes (T2DM) dramatically increase the risk of developing AD in the elderly. Both T2DM and AD share common characteristics, including inflammation, alteration of insulin signaling, insulin resistance, and glucose metabolism. Furthermore, genetic studies have demonstrated that IDE gene variations share clinical symptoms of AD as well as the risk of T2DM. The deficiency of the IDE gene may be caused either by genetic variation or by the deviation of IDE from the degradation of the amyloid-β peptide. The decreased catabolic regulation and degradation of amyloid-β peptide by IDE in favor of insulin creates an extracellular deposit and the failure of the clearance of amyloid-β peptide. Therefore, the deficiency of IDE favors extracellular deposits of amyloid-β neurotic plaques, which is one of the underlying neuropathological hallmarks of AD [181,182,183,184].

The dual role of the insulin-degrading enzyme (IDE) in degrading insulin along with amyloid-β peptide creates competition between insulin and Aβ proteins for IDE receptors; the result is in favor of insulin. Thus, the insulin cleavage mechanism prevails, because IDE is more specific to insulin and has more affinity binding sites for its receptors compared to the Aβ protein. A cross-linking study carried out by Hari et al. [185] has illustrated that insulin binds IDE specifically in the intact cells; another study carried out by Kuo et al. [186] demonstrated that the overexpression of IDE in cells in culture was found to enhance the rate of insulin degradation [173,186,187,188].

Thus, in addition to the already low amount of insulin that enters the brain due to the downregulation of BBB transporters and to the higher linkage of IDE to insulin, the free quantity of Aβ not attached to IDE is more notable and leads to Aβ accumulation in the brain, which is a hallmark of AD [189]. Moreover, insulin increases Aβ proteins in the extracellular spaces [190]. Therefore, considering the pathogenic interaction between Aβ and impaired insulin signaling, it is not surprising that central metabolic dysfunction is a certain feature of AD, illustrated by brain glucose hypometabolic changes, in addition to defects in insulin signaling, which usually proceeds AD signs and symptoms by several years [191,192]. Concerning insulin signaling consequences at the cellular level, insulin affects all the BBB’s network, including vascular endothelial cells, neurons, glial cells, and pericytes, by its involvement in the regulation of capillary vasodilatation (high concentration of insulin) and vasoconstriction (low concentration of insulin) [193,194,195]. Through this, the BBB structure and discharge of Aβ from the brain tissue into the blood vessels is maintained. However, insulin resistance negatively provokes cerebral blood pressure regulation, incremented BBB permeability, and increased intracerebral Aβ accumulation [196]. In fact, IR and hyperglycemia affect memory performance and neuronal growth, which play a role in cognitive dysfunction, a key clinical feature of AD [197].

In another way, IR has been linked to tau hyperphosphorylation tauopathy, which is a crucial pathogenic feature in AD [198]. In addition, IR affects and decreases neurotransmitters’ levels [199]. For instance, impaired insulin signaling reduces the acetylcholine level in the brain leading to crucial cholinergic perturbations, which are largely implicated in AD progression [197]. In fact, the synthesis of acetylcholine from choline and acetyl-coenzyme A (Acetyl-Co-A) is reduced significantly in AD patients [200].

CNS insulin deficiency is associated with AD pathogenesis and its progression as well [201]. In other words, AD represents a state of type 3 diabetes, where the combined effects of IR and insulin deficiency are implicated in its pathogenesis [164,201].

## 5. Pathogenic Link Between Neuroinflammation, Immune System, the BBB’s Leakiness, and Alzheimer’s Disease

Neuroinflammation is defined as a reaction to an inflammatory response within the CNS. This process of inflammation demands key pro-inflammatory mediators such as cytokines (IL-1β, IL-6, and TNFα), chemokines (CCL2, CCL5, CXCL1), secondary messengers (NO and prostaglandins), and reactive oxygen species (ROS) produced by brain glial cells (microglia and astrocytes), endothelial cells, and peripherally derived immune cells [202,203,204].

One of the mechanisms that underlies the neurodegeneration process in AD is the massive neuroinflammation [204]. Neuroinflammation plays a dual role in the brain. Firstly, it has advantageous effects and a neuroprotective role in its normally innate response when the inflammatory activity is for a shorter period, by activating the immune system, especially the phagocytic process that are presented by astrocytes and microglial cells to eliminate toxic cellular components, catabolites, and microbial pathogens. Thus [205], neuroinflammation is supposed to be a primary mechanism to conserve the homeostasis of the brain and to protect the microenvironment. Its critical role and function in protecting, saving, and restoring synaptic functions against traumatic events or contagious harm is highly substantial. Secondly, the dual role of neuroinflammation turns into a dangerous stage when a prolonged or maladaptive neuroinflammation occurs, which represents a key pathological driver for many diseases including AD, and it requires direct intervention to halt it, due to it disadvantageous effects and harmful pathological consequences for the whole brain [206,207,208]. Cytokines are the messengers during inflammation, carriers of information between different cells, and one of the substances showing pro-inflammatory activities [209]. Furthermore, they are important signaling molecules in health and disease; hence their role is critical, as neurotransmitters and hormones. Cytokines encompass chemokines, interleukins, interferons, lymphokines, and tumor necrosis factors [210].

Cytokines are synthesized mainly by macrophages and lymphocytes, as well as by polymorphonuclear leukocytes, peripheral tissues, glial cells, and other brain-resident cells. They influence various cellular functions and are involved in the regulation of systemic homeostatic functions, especially during host responses to infection. Cytokines also regulate immune system signaling, inflammation, and cell growth, survival and differentiation.

Instantly, when there is an acute inflammation within the brain, to circumvent this event, and to tackle it before it spreads throughout the brain and becomes chronic, aggressive, and intolerable, a cascade of events occurs, beginning initially with the secretion of pro-inflammatory cytokines, chemokines, small molecule messengers, tumor necrosis factors, and reactive oxygen species produced by glial brain-resident cells represented mainly by microglia. The widespread activation of this first line immune system (microglia and astrocytes) is essential to initiate the inflammatory process.

The discovery of AD risk factor genes linked to the immune response and microglia, (such as CD33 and TREM2, through GWAS) has improved our understanding of their role in the pathogenesis of AD [211,212]. For example, it has been demonstrated that TREM2 is excessively expressed in microglial cells, and it facilitates phagocytosis [213]. Further, transition to activated microglia states is associated with the upregulation of proteins like TREM2 and apolipoprotein E (APOE) [39].

The outcome of this process depends on the intensity and duration of inflammation [203]. Infiltration of peripherical immune cells into a healthy brain is not an easy task because the CNS is an immune-privileged site and has its own immune cells that strictly regulate the immune cell–BBB interaction. Indeed, glial cells represented by the innate immune system relentlessly confront strange invaders, without intervention of the peripheral immune systems. When the brain’s innate immune system is in an insolvency state or unable to combat acute, severe, and persistent inflammation, the inflammation spreads, and in this case, adaptive immune cells/peripheral immune cells and blood cells infiltrate into the brain parenchyma after the failure of the innate immune system and the compromised BBB, preventing shedding of the acquired immune cells from entering the brain, along with the main inflammatory cytokines (IL-1β, TNF-α, and IL-6) through compromised tight junctional system (TJs) of the blood–brain barrier (BBB), which further increases the BBB’s leakage and escalates neural loss. The phenomenon has largely been implicated in AD pathology [214,215].

This process is considered progressively chronic with highly destructive or pathological consequences, which in turn induces neurodegeneration and cognitive impairment [216,217,218]. The inflammatory escalation and the chronicity may occur also because of the lack of balance between anti-inflammatory and pro-inflammatory responses, due to the overactivation of microglia and cytokines [219,220]. 

The BBB normally acts as a gatekeeper of peripheral immune cells and actively contributes to immune cell trafficking into the brain. Disruption of the BBB causes it to lose its semipermeable property, which allows the trafficking of peripheral immune cells into the CNS. Isolated micro-vessels from AD patients highly express pro-inflammatory cytokines, indicating that BMECs contribute to neuroinflammation in AD [221]. Additionally, an increased number of macrophages, neutrophils, natural killer cells, T cells, and B cells that infiltrate the vessel wall or perivascular space in brain areas such as the hippocampus and frontal cortex are typically affected in AD [222,223,224,225,226,227,228,229,230,231,232,233,234,235,236]. The chronic immune response in the brain of patients with AD has been considered a substantial part of the central pathology of AD, which has been observed in AD brain autopsy and AD preclinical models. Chronic stimulation of the brain-resident microglia [237] and other immune cells has been observed, as a leading factor that triggers Aβ and tau pathologies and could be linked to the pathogenesis of AD [236]. Therefore, the brain loses this property and no longer owns the title of immuno-privileged organ, and NVU dysfunction loses this property too, due to the increased permeability and entrance of peripheral immune cells and various substances into the brain territory [39,238,239].

It is widely accepted that inappropriate activation of the innate immune system by accumulated neurotoxic materials and decreased blood flow can trigger the microglia and astrocytes to initiate an inflammatory response with the secretion of neurotoxic cytokines and chemokines. Particularly, activation of the microglia, along with the increased expression of markers associated with the innate and adaptive immune system responses, contributes to neuroinflammation and neuropathological changes in AD [240,241]. GAWS studies have calibrated risk loci that are substantially involved in the pathogenesis of AD; almost all are located near or within genes that are predominantly expressed in microglia [242]. Additionally, chronic neuroinflammation and innate immune system activation including pro-inflammatory gene polymorphisms, such as CCL3/MIP-1α and IL-6, which are produced by activated microglia, have been exhibited to be part of AD pathology and especially mediate Aβ plaques and neurofibrillary tangles (NFTs) [211]. In fact, microglial activity and the Aβ load are inversely proportional. Thus, an increase in the Aβ load causes a decrease in the activity of microglia [243,244]. Thus, the term “microgliopathy” has been suggested, which is a cogent term, recognizing that the dysfunction of microglia represents a primary disease-causing mechanism [245].

The data acquired using atomic force microscopy (AFM) or ionic strength shed light on the formation of toxic amyloid fibrils. Atomic force microscopy (AFM) was invented by Binnig et al. [246] in the mid-1980s. It is used to visualize protein molecules with high resolution [246]. AFM is used to study a variety of biological systems. It has revolutionized the field of structural biology by providing high-resolution images of proteins in their native environments, without requiring extensive sample preparation [247].

Indeed, protein aggregation is a complex biomolecular process that results from protein–protein association [248]. A possible outcome of protein aggregation is the formation of insoluble amyloid fibrils, which are often associated with the onset of the so-called conformational disorders such as Parkinson’s and AD [249].

The outcome of protein self-association is not necessarily amyloid, since it may also lead to amorphous aggregates lacking a well-defined structure. Intermediate species formed along the aggregation cascade include soluble and insoluble oligomers, protofibrils, and fibrils [250]. In the case of protein aggregation, AFM is a powerful tool, enabling the visualization and the characterization of protein aggregates with high resolution and detail [251].

Protein S100A9 is a member of the family of the so-called S100 proteins, which are calcium-binding proteins [252]. The latter are signaling molecules that have several intra- and extracellular functions, including signal transduction, cell differentiation, proteostasis, and inflammation [253]. The S100A9 protein is abundant in the brain, and increased levels of S100A9 have been found in the brains of AD patients [254], making it a robust biomarker for AD [255].

Moreover, it has been reported that S100A9 is able to rapidly form amyloid fibrils under physiological conditions in the absence of calcium [256]. Interestingly, recent results indicate that the calcium concentration is an important modulator of the S100A9 fibrillation, able to inhibit the formation of amyloid fibrils [257].

In fact, conventional AFM imaging makes it possible to identify and characterize distinct species formed at stages of protein aggregation [258,259].

Indeed, it was reported that, in the absence of calcium, S100A9 self-associates into amyloid fibrils following a two-step nucleation-autocatalytic growth model [256].

Carapeto et al. used AFM to explore the aggregation pathway of the S100A9 protein molecules in physiological conditions, in the presence of calcium at a molar ratio of 4Ca^2+^:S100A9 [260].

They found that, in these conditions, S100A9 readily forms worm-like fibrils. Furthermore, the average fibril’s length, measured as the number of periods, exhibits a moderately linear dependence on the incubation time, and the fibrils are rather flexible, able to undergo bending deformations, self-association, and assembly. Previous reports [257] showed that the presence of calcium at high concentrations (in the mM range and, therefore, well above the μM physiological threshold [261] inhibits the formation of S100A9 amyloid fibrils. In conclusion, these results suggest that, in the presence of calcium, S100A9 worm-like fibrils may follow a different aggregation pathway, presumably because they possess unique structural characteristics compared to amyloid fibrils [260,262].

Additionally, Ziaunys M et al. [262] reported that protein amyloid aggregation into insoluble fibrillar aggregates is linked to the onset and progression of multiple amyloidosis, including widespread neurodegenerative disorders such as Alzheimer‘s and Parkinson‘s diseases (PD) [263,264].

Currently, it is known that multiple environmental factors can alter both the rate and mechanism of amyloid fibril formation. They affect primary nucleation, elongation, secondary processes (surface-mediated nucleation and fragmentation), and fibril length and stability, as well as the structure of final aggregates [265]. Changes in the pH or ionic strength have been reported to affect the secondary structure/stability of the fibril [266] and their interactions with amyloid-specific compounds, such as thioflavin-T, a fluorescent amyloid probe [266], or epigallocatechin-3-gallate, an aggregation inhibitor [267].

One of the most intensely studied amyloidogenic proteins is the Parkinson‘s disease-related alpha-synuclein (α-syn) [268]. During in vitro aggregation, it has been observed on multiple occasions that different ionic strength conditions affect the rate of aggregation and can lead to the formation of distinct types of α-syn fibrils [269,270].

Salt concentration also altered their capacity to bind ThT [266], and different fibril types have even been shown to possess distinct ThT-binding properties [271].

In addition, it has recently been observed that small variations in the solution ‘s pH value can significantly alter the effectiveness of anti-amyloid compounds [272]. Considering all these factors, it seems that α-syn amyloid aggregation and the resulting structures are highly susceptible to environmental conditions, where even a small shift in certain parameters can have a major influence.

Based on these data and previous reports, it is quite clear that both ionic strength and protein concentration play a significant role in determining the type of α-syn fibrils during spontaneous aggregation.

## 6. Microglia and AD

It is appropriate to mention that microglia continuously monitor the healthy CNS, commanding the dynamic process of immunosurveillance in the CNS through multiple functions such as eliminating foreign pathogens, including bacteria, viruses, or pre-cancerous and cancerous cells from the body, providing immune defense, and maintaining homeostasis [273,274]. A meticulously controlled microglia network throughout the CNS parenchyma facilitates efficient immunosurveillance, where each cell is responsible for a special tissue territory. Each cell is recognizing and surveilling its environment and knowing the surrounding cells, dislodging cell metabolites, and maintaining tight communication with neighboring cells, facilitating cellular crosstalk.

Tissue surveillance is another central function of microglia during embryogenesis and adult CNS, with an essential role to structure, wire, and maintain neural networks [275,276]. During the healthy state, this “tissue surveillance” by microglia represents an essential process for CNS homeostasis and development [274]. The unique feature of microglia cells as housekeepers come from their possession of highly motile branching processes (amoeboid shape), and their plasticity highlights the transition between several activation states during the progression of AD pathology [277]. Initially, microglia’s function focuses on guarding and protection by contributing to tissue repair, such as Aβ clearance and combatting inflammation [278,279]. Additionally, intracerebral overproduction of Aβ and the increase in inflammatory triggers lead to continuous chronic activation of microglia during severe stages of AD [39,280,281]. This hinders the process of dislodging Aβ and increases the process of secretion of pro-inflammatory agents such as cytokines, chemokines, reactive oxygen species (ROS), and other neurotoxic products, leading to the accumulation of Aβ and thereby amplifying the general inflammatory environment that increases neuronal atrophy and synaptic disfunction [39,280,281,282,283]. Both brain-resident cells, such as neuroglial cells, and peripheral immune cells contribute to neuroinflammation. Neuroglial cells, such as microglia, have a relatively late intervention in the neuroinflammation of AD, in contrast to peripheral inflammatory factors that play a crucial role in the early stages of AD progression [284,285]. Indeed, a plethora of blood samples collected from patients with mild cognitive impairment (MCI) and AD suggest that the peripheral immune response is a very early feature of the disease [284]. The early intervention is in harmony with the activation of peripheral leukocytes and their insertion into the brain via the BBB [286,287,288,289,290]. Indeed, the trafficking of immune system cells has recently been implicated in the pathogenesis of AD, based on studies showing that neutrophils invade the brain and contribute to the induction of cognitive dysfunction and promote the neuropathology of AD [239].

## 7. Shedding Light on the Innate Immune System: Glial Cells, Mainly Microglia and Astrocytes

Glial cells are brain-resident cells and the main two players in neuroprotection and neuroinflammation, depending on the circumstances. Through development, microglia are the brain’s primary immune cells, responsible for the phagocytosis of cellular debris, and they participate in modeling the developing CNS, as well as controlling the apoptosis mechanism that occurs during early postnatal development. Furthermore, it is necessary for dislodging unnecessary synapses and removing apoptotic neurons [291,292]. The doctrine or prevailing line of thought regarding microglia as a disease indicator is now undoubted, since experimental evidence shows that both the inactivity and excessive activity of microglia are critical and dangerous. The CNS environment must favor an appropriate and specific response of microglia, with the final goal of maintaining the health of the CNS. Multifactorial risk factors (epigenetic factors and genetic variations) stimulate microglia activation in individuals exposed to environment challenges, which may provoke an aberrant microglial response that deviates from the normal neural network development. Microglia diversity in humans has been entrapped by single cells transcriptomics and moves beyond the classic concept of M1/M2 phenotypes. The classic M1-M2 dichotomy has been used traditionally to describe the microglial activation states when purified microglial cells are exposed to stimuli provoked by pathogens. Microglia, upon activation, depend on disturbances in the brain homeostasis, which can determine rapid and profound changes in microglial morphology, gene expression, and function [293,294,295,296]. Changes in gene expression, reorganization of surface molecules for interaction with the extracellular environment and neighboring cells, and the release of soluble factors acting as pro- or anti-inflammatory factors cause microglia to polarize into pro-inflammatory or anti-inflammatory phenotypes (M1 phenotype and M2 phenotype, respectively), depending on the stimulus.

The M1 type is triggered via the classical pathway by pro-inflammatory stimuli, such as interferon-γ, the lipopolysaccharide (LPS) of Gram-negative bacteria, or aggregated pathogenic proteins (Aβ, α-synuclein, and others) [297,298]. The outcome of M1 phenotype triggering is the demolition of surrounding glial and neural cells by secreting neurotoxic substances such as pro-inflammatory cytokines and chemokines like interleukin (IL)-6, tumor necrosis factor-alpha (TNF-α), C-C motif ligand (CCL)-2, superoxide, and prostaglandin-2 [299]. On other hand, the M2 phenotype functions are different, and they assist tissue repair and wound healing by secreting anti-inflammatory mediators, such as arginase-1 or chitinase-3. This phenotype can be induced by IL-4 or IL-13 in the alternative pathway or via acquired deactivation by IL-10 or the transforming growth factor-beta (TGF-β). While both phenotypes are in homeostasis during acute stimulation, the M1 phenotype (pro-inflammatory phenotype) is predominant in chronic inflammation such as AD neurodegeneration. Therefore, exaggerated microglia stimulation can lead to potentiating tissue destruction through a positive feedback loop, as occurs in almost all neurodegenerative diseases [300].

Astrocytes play a critical role in maintaining CNS function. As housekeepers, they maintain the BBB’s integrity, which in turn keeps neurotransmitters in equilibrium, and they guard synapses [301]. They also help in removing catabolites of death cells, neurofibrillary tangles, and amyloid plaques and respond to ischemia, infection, protein deposits, or other brain abnormalities via scar formation and reactive gliosis [301]. Astrocytes occupy a strategic position between capillaries and neurons. They are the most abundant cells in the brain. Astrocytic end-feet form a coating network around the brain vasculature, the glia limitans, and, together with endothelial cells and pericytes, they form the BBB, separating the bloodstream from the brain parenchyma. Astrocytes, by secreting cytokines and exacerbating mechanisms contributing to neuroinflammation, are a key player in both neurodevelopmental and neurodegenerative pathologies. Astrocytes are responsible for the maintenance of the BBB’s integrity, also producing apolipoprotein E (APOE). It was shown that APOE knock-out mice presenting with a dysfunctional BBB developed psychotic behavior, suggesting a relationship between impaired BBB function and neuropsychiatric diseases [302]. In accordance with the functions of microglia, stimulated astrocytes also display neuroprotective and neurotoxic activities. According to Liddelow et al. [301], astrocytes show different entities that are analogous to the M1/M2-type microglia, depending on the activation stimuli. The A1 astrocyte phenotype rapidly develops after acute CNS injury, such as CNS neuroinflammation. It acts immediately in response to the pro-inflammatory mediators that are secreted by the M1-type microglia, creating a secondary inflammatory response [303]. This A1-type astrocyte secretes neurotoxic factors that induce the rapid death of neurons and oligodendrocytes, thereby driving neurodegeneration and disease progression [301]. In addition, it maintains a feedback loop that galvanizes further M1-type microglia and leads to the degradation of the extracellular matrix (ECM) and tight junction (TJ) of the BBB via matrix metalloprotease (MMP) and vascular endothelial growth factor (VEGF)-A secretion [304,305].

## 8. Role of Pericytes in the Maintenance of the BBB’s Permeability

Pericytes are located within the neurovascular unit (NVU) between endothelial cells, astrocytes, and neurons. The number of pericytes involved in the barrier inversely correlates with its permeability; thus, a decrease in the number of pericytes correlates with an increase in the BBB’s permeability [306]. In addition, a reduction in pericyte coverage across the BBB is inversely correlated with aging [307] and neurodegeneration [306]. On the other hand, pericyte degeneration results in the BBB’s breakdown, with the accumulation of neurotoxic molecules leaking from the blood [308].

## 9. Existing Clinical Therapies

Despite many hypotheses and decades of debate, it is unclear whether AD is a circumscribed CNS disease or whether it is a mixture of multifactorial disorders with the involvement of risk factors both within and outside the CNS. However, regardless of the underlying causes, multidisciplinary treatment approaches are needed to manage the disease. It is well known that AD presents a formidable challenge in drug development, as reflected by a staggering 99% failure rate in clinical trials. The reason for this attrition is deeply rooted in the pathophysiology of AD and complexity of its management.

In this study, we propose to discuss possible treatment interventions that may prevent, alleviate, or slow the progression of symptoms. We have reviewed the latest therapeutic approaches in five categories: anti-amyloid therapy, anti-tau therapy, treatments that may enhance and protect the BBB’s integrity, anti-neuroinflammatory therapy, and IR diabetes mellitus.

Combination therapies that target multiple pathways may be the best option. Furthermore, a healthy lifestyle can delay the onset of AD and potentially eradicate it in many individuals. Treatments for AD include medicines that can help with symptoms and newer medicines that can help slow the decline in thinking and functioning. One available treatment is anti-amyloid therapy, in which amyloid plaques are composed of Aβ peptides in the extracellular space. Aβ is derived from the amyloid precursor protein (APP), a transmembrane protein. β-Secretase and γ-secretase cleave APP and generate pathological Aβ. The accumulation of Aβ results in neurotoxicity [309,310]. Therefore, reducing Aβ accumulation has become a therapeutic target for AD [311]. Anti-amyloid therapy comprises three strategies: secretase inhibitors, Aβ aggregation inhibitors, and Aβ immunotherapy.

Two types of medicines are currently used to treat these symptoms: cholinesterase inhibitors (Aricept, Adlarity), galantamine (Razadyne), and rivastigmine transdermal patches (Exelon). These drugs boost cell-to-cell communication. These are usually the first medicines tried, and most people experience modest improvements in their symptoms. Memantine (Namenda) works in another brain cell communication network and slows the progression of the symptoms of moderate to severe AD. It is occasionally used in combination with cholinesterase inhibitors (ChEIs). New treatments have been recently approved: ecanemab (Leqembi), which is administered as an IV infusion every two weeks, and donanemab (Kisunla), which is administered as IV infusion every four weeks for people with mild AD and MCI due to AD. Anti-tau therapy includes phosphatase modifiers, kinase inhibitors, tau aggregation inhibitors, microtubule stabilizers, and tau immunotherapy. These medications have demonstrated mild benefits and have been particularly challenging in the case of Aβ because of its structural heterogeneity and the difficulty in delivering therapeutic agents across the BBB, which are associated with severe adverse reactions, including edema and cerebral microhemorrhages. Such concerns underscore the potential need for combined therapies that simultaneously bolster BBB defenses or exploit their permeability to effectively deliver treatment. Accordingly, there has been growing interest in developing therapeutic strategies that target the BBB to enhance drug delivery, ameliorate the clearance of toxic molecules, and restore the barrier’s function. Strategies to modulate the BBB’s function in AD can be broadly categorized into three main areas: increasing the Aβ clearance across the BBB, improving the BBB’s integrity and function, and addressing neuroinflammation [24]. Accordingly, targeting neuroinflammation may provide an effective approach for maintaining a functional BBB with a reduced risk of AD or halting AD progression. Anti-neuroinflammatory strategies include microglial and astrocyte modulators, insulin resistance management, and microbiome therapy. Additionally, glucocorticosteroids, with anti-inflammatory and immunosuppressive effects, control unwanted inflammatory responses and improve the BBB’s integrity.

Therapeutic agents targeting inflammation and its downstream signaling pathways, including angiogenesis, oxidative stress, and cytoskeletal reorganization, have also been explored to restore the BBB. Moreover, age-related decreases in pericyte coverage further compromise the BBB, impairing blood flow and neuronal function. The observed upregulation of endothelial alkaline phosphatase (ALPL) in AD patients suggests that it is a potential therapeutic target, as its inhibition could modify the BBB’s permeability and influence disease progression. The molecular links between dysregulated insulin signaling in AD and diabetes raise the prospect of novel therapeutic strategies for AD, based on antidiabetic agents. Intranasal insulin therapy has been used to treat AD. Intranasal insulin administration with intranasal regular insulin at 40 IU daily for 4 months improved memory impairment and verbal memory in MCI and AD [309] and, importantly, improved performance in patients with early AD.

The protective role of insulin thus derives from IR signaling-dependent downregulation of oligomer-binding sites in neuronal processes, indicating the occurrence of cellular mechanisms that physiologically protect synapses against Aβ oligomers [309].

However, at later disease stages, when surface IRs dwindle, insulin may stimulate other receptors (e.g., IGF-1R), thus improving AD-related deficits. Nevertheless, alternative approaches to bypass IRs include glucagon-like peptide-1 receptor (GLP-1R) agonists, which are an attractive option because they activate pathways common to insulin signaling through G-protein-dependent signaling [310]. GLP-1Rs are present in cultured neurons as well as in rodent and human brains. Exendin-4 and liraglutide are GLP-1R agonists that have recently been approved for the treatment of type 2 diabetes. Exendin-4 and liraglutide also restored impaired insulin signaling in the brains of a transgenic mouse model of AD, improving cognition and decreasing Aβ accumulation [311]. As recently suggested, an agent that chronically decreases Aβ levels would be beneficial in APOE ε4 carriers [311].

Other pharmacological interventions aimed at preserving mitochondrial biogenesis-integral functionality and mitophagy of diseased organelles may attenuate the adjacent spillover of free radicals that further perpetuate mitochondrial damage and catalyze inflammation. Alternative medicines, such as herbal remedies, vitamins, and other supplements, such as vitamin E, omega-3 fatty acids, curcumin, ginkgo, and melatonin, are widely used to promote cognitive health and prevent or delay Alzheimer’s disease. However, clinical trials have yielded inconsistent results. There is little evidence to support their use as effective treatments. Lifestyle and home remedies were also considered (exercise, nutrition, and social engagement and activities). In fact, healthy lifestyle choices promoted good overall health. They may also play a role in maintaining brain health.

## 10. Conclusions

A plethora of hypotheses, ranging from tauopathy, cholinergic hypothesis, neuroinflammation, amyloidogenic cascade, and oxidative stress to the disruption of the BBB have been suggested to explain the pathogenesis of AD. There is, however, no consensus and there are no convincing data to explain its phenomenology [85,181,312,313,314,315]. However, it appears that multiple factors orchestrate together to cause this devastating illness. A combination of age-related changes and genetic, environmental, and lifestyle factors may be the underlying cause of AD. These factors may lead to the accumulation of plaques (Aβ) and neurofibrillary tangles (tau). Other biological processes have also emerged in its pathogenesis in recent years. These processes include the breakdown and/or dysfunction of the blood–brain barrier (BBB); carriers of the gene variant APOE4 and its link to BBB; and neuroinflammation, in addition to type 2 diabetes (T2DM), metabolic syndrome (MS), and brain insulin resistance (IR). Even though it is unclear whether the BBB is the cause or consequence of AD, its disruption in AD has been extensively reported [70,316]. It is implicated in the initiation and development of AD, particularly because BBB damage promotes the buildup of Aβ toxins in the brain [72,317]. Thus, the BBB’s breakdown should be considered an important pathophysiological factor in AD [318].

BBB dysfunction could lead to a chain of events in neurodegenerative disorders, including increased BBB permeability, microbleeds, impaired glucose transport, impaired P-glycoprotein function, perivascular deposits, and the accumulation of amyloid-β (Aβ), especially in AD pathology, because the BBB is what removes Aβ across the barrier. Thus, decreased clearance abilities may therefore promote the built-up of Aβ plaques, which is an initial insult sufficient to initiate neuronal loss and neurodegeneration [24,25,26,27,319].

BBB damage may also induce neurodegenerative processes via the activation of inflammatory pathways and via the penetration of neurotoxic blood-derived products, pathogens, and cells across BBB into the brain [28]; these primary malefactors that cause brain damage, shortly after BBB dysfunction, are associated with inflammatory and immune responses [29] and provoke pericyte-mediated cerebral hypoperfusion [30], which altogether can initiate multiple pathways of neurodegeneration. Undoubtedly, the accumulation of neurotoxic material and hypoperfusion can activate glial cells (astrocytes and microglia within the brain), leading to an inflammatory response with the secretion of chemokines and cytokines [320].

It is well-known that the innate immune system is a brain safeguard in health and disease; it is the first front in confronting any abnormal event in the brain, and it is primarily engaged in neuroinflammation in AD. Strikingly, activated astrocytes and microglia around plaques have shown by different studies; they release pro-inflammatory materials and trigger further inflammatory processes. Thus, glial cells-mediated inflammation holds two standards: advantageous and disadvantageous or a “double-edged sword role”, causing both beneficial and harmful effects in AD. Indeed, the chronic activation of microglia in the brain, aside from other immune cells, exacerbates Aβ and tau pathologies and could be a link in the pathogenesis of AD [321,322,323].

In addition, the leakage of the BBB’s tight junctions (TJs) increases the permeability of the BBB; this process paves the way for the infiltration of peripheral macrophages and neutrophils into the brain and activates the innate immune response. In addition to the entry of circulating leukocytes into the brain and the influx of T and B lymphocytes, this action indicates that both the adaptive and innate immune systems orchestrate together, which may cause catastrophic consequences and affect badly the integrity of the brain parenchyma, leading to more neural damage [24,72,324].

The importance of APOE in physiology and disease is well-known. Its role is essential for the normal catabolism of triglyceride-rich lipoprotein constituents. In the CNS, APOE is expressed mainly in astrocytes and microglia, and in the peripheral tissues, it encodes a major lipid-carrier protein in the brain [77], as well as vascular mural cells and choroid plexus cells. APOE modulates multiple pathways; its activities are associated with the endocytosis of lipoproteins, synaptic plasticity, membrane integrity, neurogenesis and neuronal degeneration, neuroinflammation, mitochondrial function, tau phosphorylation, and Aβ metabolism [78].

Moreover, APOE has a crucial role in amyloid-beta-protein (Aβ) clearance, aggregation, and deposition [82,83]. The main associated pathological isoform of APOE in AD is the APOE-ε4 genotype, which is the highest risk category for late-onset Alzheimer’s disease (LOAD), with the underlying mechanism of this link being both presynaptic and postsynaptic dysfunction [84]. The APOE ε4 gene variant promotes Aβ plaque formation [85], which facilitates the loss of key presynaptic proteins [86], disrupts long-term potentiation and plasticity [87], and leads to a reduction in dendritic density [88].

In this context, it was shown that individuals who were cognitively intact and carried either one or two copies of APOE4 had a leaky BBB, initially in the hippocampus and in the para-hippocampal gyrus [93]. Remarkable atrophy in these two regions, due to BBB dysfunction was observed, leading to memory and cognitive impairment. Moreover, histological analysis of post-mortem brain tissue has reported that the BBB’s breakdown in AD patients reduced the cerebral blood flow and caused neural loss and behavioral deficits independent of Aβ, and it was more noticeable among APOE4 carriers compared to those with APOE3 or APOE2 [94,95,96,97,98].

APOE4 activates an array of proteins, beginning with the protein cyclophilin A (CypA) in the pericytes, which, in turn, stimulates a downstream signaling pathway involving activation of the inflammatory protein matrix metalloproteinase-9 (MMP9) in pericytes and in endothelial cells. This activation by APOE4 leads to the MMP-9–mediated degradation of the BBB’s tight junction and basement membrane proteins causing the BBB’s breakdown [93,99,100,101].

In addition, cyclophilin A (CypA) and matrix metalloproteinase-9 MMP9) proteins, which are part of the inflammatory pathway implicated in APOE4-driven pericyte damage and BBB breakdown, were all are elevated in the CSF of APOE4 carriers [98,104]. Thus, APOE4-status is a risk factor for the BBB’s breakdown via activation of the Cyp-A-MMP9 pathway [99,105] and has been associated with increased hippocampal BBB leakiness and higher sPDGFRβ [93], the novel and sensitive biomarker of BBB disfunction [106,107].

Another player involved in the pathogenesis of AD is IR. AD–IR’s common pathologic exclusiveness encompasses amyloid genesis, bioenergetic dysfunction, inflammation, and obesity, which together strengthen the notion that insulin resistance may accelerate the appearance of AD [149,150]. One of the mechanisms by which insulin impacts cognitive abilities is by affecting the cerebral energy metabolism. IR has catastrophic consequences on brain function [149,150]. Definitively, IR is observed among individuals with impaired insulin-stimulated glucose output into adipocyte tissues and muscle, accompanied by impaired insulin suppression of hepatic glucose output [156]. This phenomenon of reduced cells’ response to insulin leads to hyperinsulinemia, which occurs due to genetic polymorphisms; tyrosine phosphorylation of the insulin receptor, insulin receptor proteins, PIP-3 kinase; or abnormalities of GLUT 4 function; and/or environmental factors [157,158,159,160].

Insulin resistance is a complicated pathophysiological disorder with impaired biologic response of target tissues to insulin stimulation, impaired ability to inhibit glucose production and stimulate peripheral glucose elimination, and often, hyperinsulinemia to maintain normoglycemia [161].

IR etiology depends on any factor causing disturbances in the insulin signaling pathway in the host, including decreased peripheral target tissue responsiveness to insulin, abnormalities in receptor binding, autophagy, and intestinal microecology, as well as metabolic dysfunction of the liver and other abnormalities in the host extracellular environment such as lipo-toxicity, inflammation, hypoxia, and immunity abnormalities that can trigger intracellular stress factors in key metabolic target tissues, which impairs the normal metabolic activity of insulin in these tissues thereby provoking the progression of whole-body IR [162,163].

A plethora of pathways were suggested to explain the link between AD and IR [154,155]. Initially, in case of the insulin resistance state, several detrimental events occur: IR, followed by compensated peripheral hyperinsulinemia and the resultant hyperglycemia or glucose intolerance. When IR develops, a compensatory hyperinsulinemia occurs, due to the increased secretion of insulin (extra-insulin) from the pancreatic β-cell to achieve normoglycemia, which leads to an inadequate or vicious cycle of IR ↔ hyperinsulinemia [154,155,158,164,165,166].

This detrimental cycle of IR–hyperinsulinemia causes metabolic consequences including hyperglycemia, high blood pressure, hyperuricemia, dyslipidemia, high levels of elevated inflammatory markers, endothelial dysfunction, and cardiovascular diseases and may lead to metabolic syndrome and type 2 diabetes. Together, these consequences may be implicated in AD pathogenesis to different degrees [158,165,166,167].

The chronic elevation in peripheral insulin (peripheral hyperinsulinemia) levels impacts central insulin availability and function. Indeed, peripheral hyperinsulinemia leads to an increase in the insulin level in the brain, because the transport of molecules across the BBB is highly affected by the variation in their peripheral levels, especially the high level of insulin [149,167,168].

Insulin is degraded into the brain by the insulin-degrading enzyme (IDE), also named-insulysin); structurally, in humans, the gene encoding IDE is located on the long arm of chromosome 10 (q23-q25) and contain 24 exons and a large sequence of introns [169].

IDE, originally known as the main enzyme involved in the cleavage of insulin as well as other amyloidogenic peptides, such as the β-amyloid (Aβ) peptide, eliminates Aβ’s neurotoxic effects, one of the hallmarks of Alzheimer’s disease (AD), which shows the relationship between IDE, diabetes, and AD [171,172].

Thus, IDE has been long envisaged as a potential therapeutic option in metabolic and neurodegenerative diseases [173].

However, the IDE cleaves numerous peptides unevenly, including β-amyloid, demonstrating a critical role in the pathophysiological processes regulated by these peptides [174,175,176,177,178,179]. It is well-known that IDE represents the link and the key factor in the crosstalk between hyperinsulinemia and AD [180]. Furthermore, genetic studies have demonstrated that IDE gene variations share clinical symptoms of AD as well as the risk for type 2 diabetes (T2DM). An optional explanation for the deficiency of the IDE gene is that it may be caused by either genetic variation or by the deviation of IDE from the degradation of amyloid-β peptide. Indeed, decreased catabolic regulation and degradation of amyloid-β peptide by IDE in favor of insulin creates an extracellular deposit and the failure of the clearance of amyloid-β. Therefore, the deficiency of IDE favors extracellular deposits of amyloid-β neurotic plaques, which is one of the underlying neuropathological hallmarks of AD [181,182,183,184].

The dual role of the insulin-degrading enzyme (IDE) in degrading insulin along with the amyloid-β peptide creates competition between insulin and Aβ protein for IDE receptors, the result favors insulin. Thus, the insulin cleavage mechanism prevails, because IDE is more specific to insulin and has more affinity binding sites for its receptors compared to the Aβ protein.

Thus, in addition to the already low amount of insulin that enters to the brain due to the downregulation of BBB transporters and to the higher linkage of IDE to insulin, the free quantity of Aβ not attached to IDE is more notable and leads to Aβ accumulation in the brain, which is one of the important hallmarks of AD [189]. Therefore, considering the pathogenic interaction between Aβ and impaired insulin signaling, it is not surprising that central metabolic dysfunction is a certain feature of AD, illustrated by brain glucose hypometabolic changes, in addition to defects in insulin signaling, which usually proceeds AD signs and symptoms by several years [191,192]. Concerning insulin signaling consequences at the cellular level, insulin affects the entire BBB network, including vascular endothelial cells, neurons, glial cells, and pericytes, by its involvement in the regulation of capillary vasodilatation (high concentration of insulin) and vasoconstriction (low concentration of insulin) [193,194,195]. Through this, the BBB’s structure and discharge of Aβ from the brain tissue into the blood vessels is maintained. However, insulin resistance negatively impacts the cerebral blood pressure regulation, increases the BBB’s permeability, and increases intracerebral Aβ accumulation [196]. In fact, IR, as well as hyperglycemia, affects memory performance and neuronal growth, which plays a role in cognitive dysfunction, a key clinical feature of AD [197].

In another way, IR has been linked to tau hyperphosphorylation tauopathy, which is a crucial pathogenic feature in AD [198]. In addition, IR affects and decreases neurotransmitters’ levels [199]. For instance, impaired insulin signaling reduces the acetylcholine level in the brain, leading to crucial cholinergic perturbations, which are largely implicated in AD progression [197]. In fact, the synthesis of acetylcholine from choline and acetyl-coenzyme A (Acetyl-Co-A) is reduced significantly in AD patients [200].

## 11. Summary

Many hypotheses have investigated the conditions that undermine cognitive functioning. However, the basic pathogenesis of AD is still debated. Great efforts are being made to better comprehend what enables, causes, or worsens AD. Many researchers have focused on neuroinflammation, the dysfunction of the BBB, the link between AD and IR, and the link between the apolipoprotein epsilon-4 allele and BBB and AD (Figure 1, Figure 2 and Figure 3). Neuroinflammation is a two-edged sword, a well-intentioned but sometimes destructive helper, which has been studied extensively. Understanding in more depth how immune reactions interact with the various features of AD increases our efforts to determine the prevention and appropriate treatment of chronic inflammatory conditions by blocking the inflammatory proteins that microglia release when activated. Additionally, the BBB is a highly selective semipermeable structural and biochemical barrier, which ensures the stable internal environment of the brain and prevents foreign objects from invading the brain tissue. The BBB is critical for brain Aβ homeostasis and regulates Aβ transport. Faulty BBB clearance of Aβ through deregulated LRP1/RAGE-mediated transport, aberrant angiogenesis, and arterial dysfunction may initiate neurovascular uncoupling, Aβ accumulation, cerebrovascular regression, brain hypoperfusion, and neurovascular inflammation. Indeed, the BBB’s breakdown has been suggested as an early marker for AD; yet, the relationship between the BBB’s breakdown and AD-specific biomarkers based on the amyloid/tau/neurodegeneration framework still needs more clarification [319].

The importance of a healthy BBB for therapeutic drug delivery and the adverse effects of disease-initiated pathological BBB breakdown in relation to the brain delivery of neuropharmaceuticals are extremely important. The characterization of molecular mechanisms controlling vascular inflammation and leukocyte trafficking could therefore help to determine the basis of the BBB’s dysfunction during AD and may lead to the development of new therapeutic approaches. In fact, the need for researching future directions, addressing the gaps in the field, and taking the opportunities to control the course of neurological diseases by targeting the BBB are warranted [325]. 

Furthermore, glial cells, which are normally responsible for maintaining the homeostasis of synaptic transmission and its remodeling by pruning, are the initiators of neuroinflammation and toxic tau and amyloid-β (Aβ) accumulation. Thus, they deliver the brain into a situation of sustained or even self-accelerated deterioration. We explain their function and their role in the neuroinflammation, the cell types and mediators involved in neuroinflammation and AD, the symptom manifestations in clinical settings, and potential candidates for improving diagnosis and treatment [326,327,328].

Insulin resistance (IR) is a complicated pathophysiological disorder with the impaired biologic response of target tissues to insulin stimulation and the impaired ability to inhibit glucose production and stimulate peripheral glucose elimination, which often comes with hyperinsulinemia to maintain normoglycemia [161].

IR etiology depends on any factor causing disturbances in the insulin signaling pathway in the host, including the decreased peripheral target tissue responsiveness to insulin, abnormalities in receptor binding, autophagy, and intestinal microecology, as well as metabolic dysfunction of the liver and other abnormalities in the host extracellular environment such as lipo-toxicity, inflammation, hypoxia, and immunity abnormalities that can trigger intracellular stress factors in key metabolic target tissues, which impairs the normal metabolic activity of insulin in these tissues, thereby provoking the progression of whole-body IR [162,163].

A plethora of pathways were suggested to explain the link between AD and IR [154,155]. Initially, in the case of an insulin resistance state, several detrimental events occur: IR, followed by compensated peripheral hyperinsulinemia and the resultant hyperglycemia or glucose intolerance.

The chronic elevation in peripheral insulin (peripheral hyperinsulinemia) levels impacts central insulin availability and function. Indeed, peripheral hyperinsulinemia leads to an increase in the insulin level in the brain, because the transport of molecules across the BBB is highly affected by the variation in their peripheral levels, especially the high level of insulin [149,167,168].

Insulin is degraded into the brain by the insulin-degrading enzyme (IDE), originally known as the main enzyme involved in the cleavage of insulin as well as other amyloidogenic peptides, such as the β-amyloid (Aβ) peptide, and it eliminates Aβ’s neurotoxic effects—one of the hallmarks of Alzheimer’s disease (AD)—which shows the relationship between IDE, diabetes, and AD [171,172].

It is well known that IDE represents the link and the key factor in the crosstalk between hyperinsulinemia and AD [171]. Furthermore, genetic studies have demonstrated that IDE gene variations share clinical symptoms of AD as well as the risk for type 2 diabetes (T2DM). An optional explanation for the deficiency of IDE gene may be either genetic variation or the deviation of IDE from the degradation of amyloid-β peptide. Indeed, decreased catabolic regulation and degradation of amyloid-β peptide by IDE in favor of insulin creates an extracellular deposit and the failure of the clearance of amyloid-β. Therefore, the deficiency of IDE favors extracellular deposits of amyloid-β neurotic plaques, which is one of the underlying neuropathological hallmarks of AD [181,184].

The dual role of the insulin-degrading enzyme (IDE) in degrading insulin along with amyloid-β peptide creates competition between insulin and Aβ protein for IDE receptors; the result is in favor of insulin. Thus, the insulin cleavage mechanism prevails, because IDE is more specific to insulin and has more affinity binding sites for its receptors compared to the Aβ protein. Thus, IDE has been long envisaged as a potential therapeutic option in metabolic and neurodegenerative diseases [173].

## Figures and Tables

**Figure 1 ijms-26-08253-f001:**
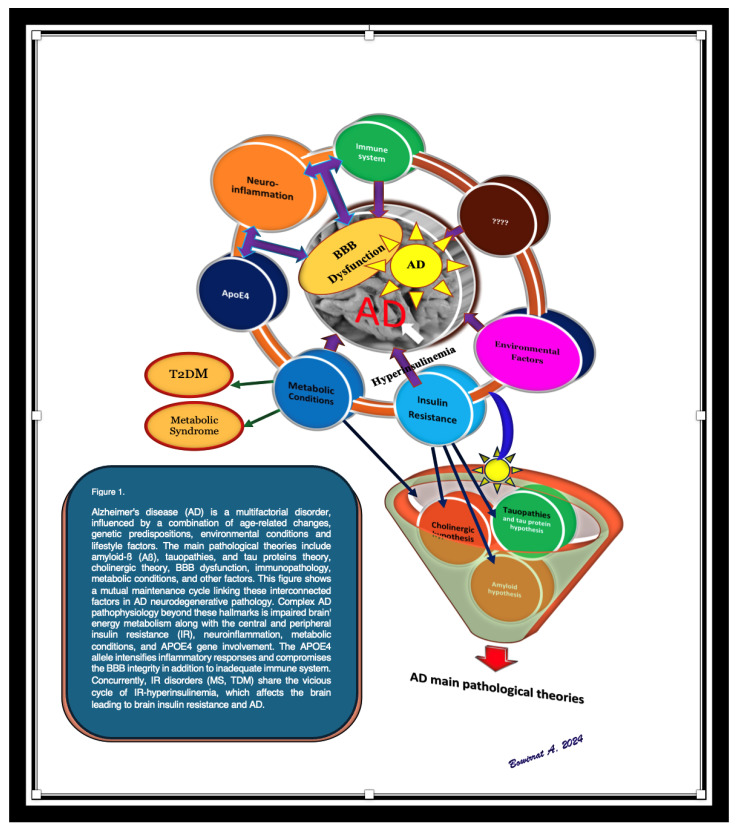
Different pathologies involved in Alzheimer’s disease.

**Figure 2 ijms-26-08253-f002:**
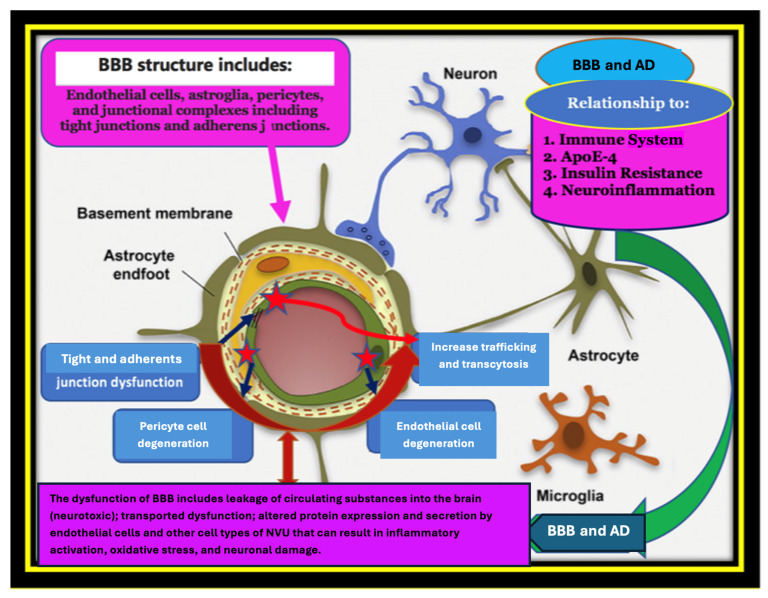
Alzheimer’s Disease (AD) is a multifactorial disorder, caused by a combination of age-related changes and genetic, environmental, and lifestyle factors that affect the brain over time. A plethora of substantial and biological processes have emerged in terms of its pathogenesis such as the breakdown and/or dysfunction of the blood–brain barrier (BBB); patients who are carriers of the gene variant APOE4; and the immune-senescence of the immune system. Furthermore, type 2 diabetes (T2DM) and metabolic syndrome (MS) have also been observed as early markers that may provoke pathogenic pathways that lead to or aggravate AD progression and pathology. There are numerous AD features that require further understanding, such as chronic neuroinflammation, decreased glucose utilization and energy metabolism, as well as brain insulin resistance (IR). In this figure, we describe the physiological and the pathological in terms of the blood–brain barrier (BBB) and its relationship with AD. We tried to connect the dots of the multiple comorbidities and their cumulative synergistic effects on BBB dysfunction and AD pathology. We shed light on the path-physiological modifications in the cerebral vasculature that may contribute to AD pathology and cognitive decline prior to clinically detectable changes in amyloid-beta (Aβ) and tau pathology, diagnostic biomarkers of AD, neuroimmune involvement, and the role of the APOE4 allele and AD–IR pathogenic link—the shared genetics and metabolomics biomarkers between AD and IR disorders.

**Figure 3 ijms-26-08253-f003:**
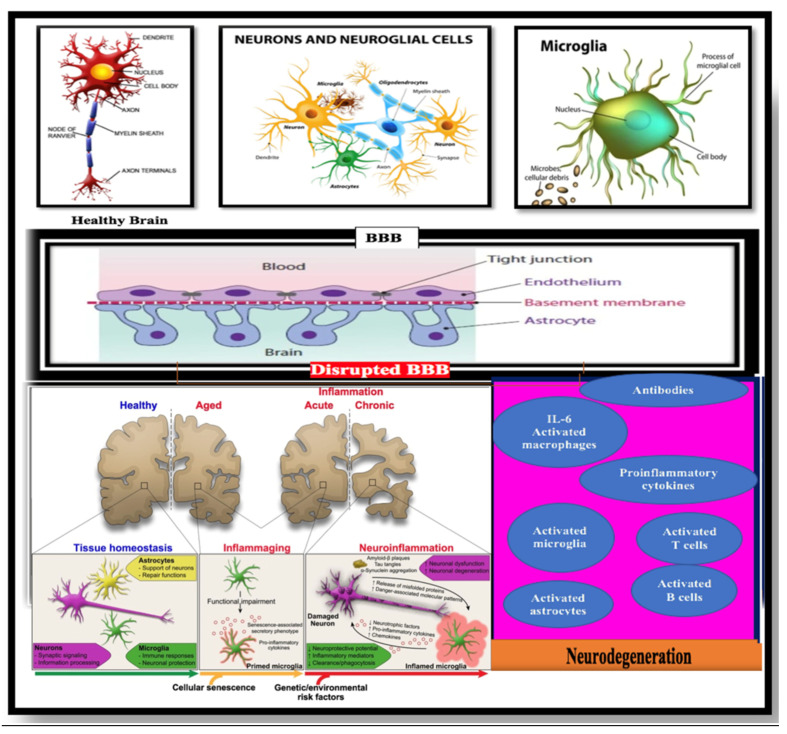
Relationship between the central nervous system (CNS) and systemic immune responses in Alzheimer’s disease (AD) patients (Bowirrat A, 2022) [35].

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
