# Peer review of "Navigation Between Alzheimer’s Disease (AD) and Its Various Pathophysiological Trajectories: The Pathogenic Link to Neuroimmunology—Genetics and Neuroinflammation"

_ijms, 2025, doi:10.3390/ijms26178253_

Round 1

Reviewer 1 Report

Comments and Suggestions for Authors

The manuscript titled “A Navigation Between Alzheimer’s Disease (AD) and its Various Pathophysiological Trajectories: The Pathogenic link to Neuroimmunology; Genetics and Neuroinflammation” by Bowirrat, A.; et al. is a scientific work where the authors outlined the most recent advances in the field of Alzheimer’s disease of those factors that lead to the onset and progression of this neurodegeneration malignancy. This is a topic of growing interest and the manuscript is generally well-written.

However, it exists some points that need to be addressed (please, see them below detailed point-by-point) to improve the scientific quality of the submitted manuscript paper before this article will be consider for its publication in Frontiers in Nanotechnology.

1) Introduction. “Alzheimer’s disease (AD) is an aging complex neuro-degenerative brain pathology (…) AD is a predominant and incurable chronic debilitating disorder occupying more than 60%-80% of all types of dementias” (lines 49-56). Could the authors provide quantitative data insights according to the worldwide global incidence burdens of Alzheimer’s diseases and the related disability-adjusted life years (DALYs)? This will significantly aid the potential readers to better understand the significance of this devoted research.

2) “2. Pathogenic Link Between Blood-Brain Barrier (BBB), APOE4 Polymorphism and Alzheimer’s Disease (AD)” (lines 92-201). A schematic representation will also benefit the potential readers to better visualize the hidden underlying mechanisms among Alzheimer’s Disease, the blood-brain barrier and the APOE4. Similar comment for the section “4. Pathogenic Link Between Insulin Resistance (IR), Alzheimer’s Disease (AD), and Blood-Brain Barrier (BBB) disorders” (lines 307-508) and “5. Pathogenic Link Between Neuroinflammation, Immune System, BBB leakiness and Alzheimer’s Disease” (lines 509-612). The Figure 1 (line 986) is fine to have a more complete outlook, but specific figures for each sections are also requested.

3) “IDE (…) Zn2+” (lines 451-452). As minor remark, the cation valence needs to appear as superscript.

4) “Additionally, chronic neuroinflammation and innate immune system (…) microgila have been exhibited to be part of AD pathology and especialy mediate Aβ plaques and neurofibrillary tangles (NFTs)” (lines 603-607). Here, even if I agree with the information detailed in these statements by the authors, it may be also advisable to discuss how the presence of certain positive divalent cations [1] or the ionic strength [2] can triggers the formation of toxic amyloid fibrils. This will strengthen the outcomes found in this field.

[1] https://doi.org/10.3390/biom14091091

[2] https://doi.org/10.3390/ijms222212382

5) “In fact, the relationship between microglia activity and the (…)” (lines 607-608). The lettering size should be homogenized with the rest of the content. This comment needs to be taken into account for the rest of the main manuscript body text.

6) Finally, the authors should highlight some recently reported existing clinical therapies to fight against neurodegeneration malignancies.

7) “10. Summary” (lines 909-984). This section perfectly remarks the most relevant outcomes found by the authors in this field and also the promising future prospectives. It may be desirable to add a brief statement to discuss about the potential future action lines to pursue the topic covered in this work.

Author Response

we did the necessary changes according to the reviewer suggestions

Reviewer 2 Report

Comments and Suggestions for Authors

The authors intend to review the contribution of neuroinflammation, dysfunction of BBB, insuline resistence and apolipoprotein polymorphism to the development and progression of Alzheimer disease (AD).

Unfortunately, although the manuscript contains a substantial amount of information, it is not presented in a clear and useful manner. The structure of the manuscript should be optimized in order to improve the impact and potential application of the content. The paper would benefit from a reorganization of each section into two distinct parts: one focusing on the anatomical and physiological functions in healthy individuals, and the other on the participation/contribution of the respective factors to the pathophysiology of AD.

There is noticeable overlap of information across sections, particularly between the Conclusion and Summary sections. It is expected to have correspondence between the title and its content. For example, although the APOE4 polymorphism is mentioned in the title of Section 2, it is primarily discussed in Section 3.

The authors should avoid using different abbreviations for the same terms. It is an usual practice also to follow the rule that abbreviation in the text appears after the first mention. Additionally, all sentences should begin with capital letters.

Comments on the Quality of English Language

The English needs substantial improvement. The use of shorter and more clearly formulated sentences would be beneficial to the text's quality.

Author Response

thank the reviewer for his comments and we did the necessary changes

Round 2

Reviewer 1 Report

Comments and Suggestions for Authors

The authors did a great deal of effort to cover all the suggestions raised by the Reviewers and for this reason, the scientific manuscript quality was greatly improved

Author Response

Thank you very much for your comments and inputs we agree with you completely. we did all the necessary changes  

Reviewer 2 Report

Comments and Suggestions for Authors

It appears that the authors have misunderstood the suggestion about the structure of sections. The point is to follow the same structure across the sections. There is no need to "write a book", it is enough to reorganize the already existing information in the same way as the authors have already done in Sections 2 and 3.

Different abbreviations for the same terms are still present in the manuscript:

- line 147 (BMVECs) and line 156 (BMECs)

- line 222 (ApoE) and line 225 (APOE)

Comments on the Quality of English Language

Since good-quality English is essential for readers, the manuscript should be revised - grammaticaly and lexically.

The paragraphs  between 75 and 79, 83 and 89, 90 and 93 lines are only few examples that need correction.

The sentences on lines 166, 173 need capital letters. The sentence on line 193 does not represent a complete thought.

Author Response

We agree with the reviewer report. 

1) first comments. we agree

answer: we change the structure of sections according to the reviewer suggestion, we colored them in Red color: as sections 2, 3, 4, 5...

2)  comment 2: different abbreviations for the same terms: 

answer: We agree and we did similar abbreviations 

3) English language

answer: we did again and again editing and we fix any language problem or grammar in Red color 

4) paragraphs  between 75-79, 83-89, 90-93 

answer: changes have been made accordingly within the text in RED color

5) comments sentences on lines 166, 173. need capital letters

answer: we change the to capital letter